# Recommendation Method of Power Knowledge Retrieval Based on Graph Neural Network

**Rongxu Hou [1], Yiying Zhang [2],\*, Qinghai Ou [3], Siwei Li [3], Yeshen He [4], Hongjiang Wang [1] and Zhenliu Zhou [1]**

[1] College of Information Shenyang Institute of Engineeringy, Shenyang 110135, China; wanghj@sie.edu.cn (H.W.); zhouzl@sie.edu.cn (Z.Z.)
[2] College of Artificial Intelligence, Tianjin University of Science & Technology, Tianjin 300457, China
[3] Beijing Zhongdian Feihua Communication Co., Ltd., Beijing 100070, China; ouqinghai@sgitg.sgcc.com.cn (Q.O.); lisiwei@sgitg.sgcc.com.cn (S.L.)
[4] China Cridcom Co., Ltd., Shenzhen 518109, China; heyeshen@sgchip.sgcc.com.cn
\* Correspondence: yiyingzhang@tust.edu.cn

**Abstract:** With the development of the digital and intelligent transformation of the power grid, the structure and operation and maintenance technology of the power grid are constantly updated, which leads to problems such as difficulties in information acquisition and screening. Therefore, we propose a recommendation method for power knowledge retrieval based on a graph neural network (RPKR-GNN). The method first uses a graph neural network to learn the network structure information of the power fault knowledge graph and realize the deep semantic embedding of power entities and relations. After this, it fuses the power knowledge graph paths to mine the potential power entity relationships and completes the power fault knowledge graph through knowledge inference. At the same time, we combine the user retrieval behavior features for knowledge aggregation to form a personal subgraph, and we analyze the user retrieval subgraph by matching the similarity of retrieval keyword features. Finally, we form a fusion subgraph based on the subgraph topology and reorder the entities of the subgraph to generate a recommendation list for the target users for the prediction of user retrieval intention. Through experimental comparison with various classical models, the results show that the models have a certain generalization ability in knowledge inference. The method performs well in terms of the MR and Hit@10 indexes on each dataset, and the F1 value can reach 87.3 in the retrieval recommendation effect, which effectively enhances the automated operation and maintenance capability of the power system.

**Keywords:** graph neural network; knowledge inference; search intent prediction; recommender system

## 1. Introduction

With the rapid development of computational science and information technology, artificial intelligence is deeply integrated with various industries, which greatly promotes the digitalization processes of national economic industries [1]. While the continuous improvement and development of digital technology in power grids has promoted the efficient operation and automation of power systems, the constant iterative updating of technical equipment, expert cases, and standard requirements also lead to the rapid growth of power standard data while information is instantly sensed and highly shared. This has led to a series of problems while enabling rapid information sharing [2]. Due to the scattered and excessive number of documented materials of power grid standards and the explosive growth of data due to real-time information sensing and sharing, this leads to the poor matching of traditional search results and difficulties in information screening. At the same time, the complexity of the stored data structure leads to the low real-time efficiency of retrieval and the existence of low information visualization [3]. Especially in fault retrieval in the power domain, there are often complex topological connections between grid devices,

and it is increasingly difficult to correctly obtain the user's retrieval needs [4]. Therefore, mitigating the effects of power information overload and accurately helping users to match results that meet their individual needs are the main research objectives of this study.

Pioneered by Google and applied to search engines, the knowledge graph expresses knowledge as attribute–value pairs corresponding to entities and represents the relationships between knowledge through connections between entities [5]. Its data sources are mainly web data, such as encyclopedic pages like Wikipedia and databases in the industry. It can alleviate problems such as data sparsity because of its ability to mine potential semantic associations through link prediction between entities [6]. Meanwhile, the knowledge graph embedding method can embed entities and relationships into a low-dimensional vector space and retain the structural information of the knowledge graph, which can effectively improve the recommendation quality [7,8]. Based on the unique storage structure of knowledge graphs, graph neural networks are introduced to recommender systems due to their good graphical representation learning ability [9,10].

Retrieval recommendation systems are used to match and recommend relevant items that satisfy the user's intent during a search query. One of the most representative approaches is collaborative filtering, which uses historical user interactions to make recommendations based on users' common preferences [11,12]. However, such methods can hardly avoid the cold-start problem that exists due to the sparse interaction data [13]. Current research on recommendation methods is mainly focused on modeling product and user representations [14]. Generally, text-based or relationship-based approaches are mostly used to construct behavioral models for web logs to mine user retrieval intent [15,16]. Meanwhile, the search ranking algorithm is improved based on user intent, using semantic relevance to analyze the user query intent and reordering matching based on the query results [17,18].

Although retrieval recommendation systems are now better able to achieve personalized user perceptions and the efficient and accurate ranking of recommendation results, there are still some problems in the diversification and inference of recommendation effects when targeting different domains and audiences [19]. In terms of fine-grained inference, only the characteristics of items are considered, relying on the items' representational information judgment, failing to derive their relevant fine-grained features. Meanwhile, machine learning algorithms require more manual annotation and the framework is often time-sensitive and needs to take into account the retrieval features in the scenario of cold-start and regular retrieval [20]. The current system does not fully utilize the topological information of the knowledge graph, and the mining of aspects such as relational reasoning is not deep enough. Therefore, it is difficult to achieve the accurate prediction of users' retrieval interests [21].

In order to solve the problems of cold starts in retrieval methods and the existence of information overload in the recommended results, we propose a retrieval recommendation method for electric power knowledge based on the current research of recommendation technology. Our main contributions are as follows.

- Making full use of the network topology of the knowledge graph, we adopt a graph neural network to realize the deep semantic embedding and knowledge inference of electric power knowledge. The completeness of the power knowledge map is improved by complementing the potential entity relationships of the existing knowledge map.
- We deeply mine the features of users' retrieval behaviors and analyze users' personality subgraphs and retrieval subgraphs through knowledge aggregation and similarity matching. Based on the path and topology of the subgraph, we reorder the power entities to achieve the accurate prediction of users' retrieval intentions and avoid the problems of cold starts and information overload.
- We design comparison experiments on public datasets to verify the recommendation effectiveness of the graph-neural-network-based user retrieval recommendation method, demonstrate the recommendation basis of the model results, and enhance the interpretability of the algorithm.

This method enables the faster and more accurate recommendation of knowledge and cases for grid practitioners with different positions and needs and improves the safety of the grid and the timeliness of troubleshooting. We introduce the current status of domestic and international research on recommendation systems in Section 2, the graph-neural-network-based recommendation method for power knowledge retrieval in Section 3, the experimental analysis and comparison in Section 4, and the conclusions and remaining problems of this method in Section 5.

## 2. Related Work

The research of retrieval recommendation systems has been well developed, especially in the big data information environment, such as user interest perception and entity feature representation, which can allow users to obtain more personalized retrieval results. The following is a description of the existing recommendation methods, which are still in the development stage in terms of combining data storage features with knowledge graphs.

In recent years, recommendation system research has gradually turned to deep learning and reinforcement learning in the form of research based on the original keyword and ranking models [18,19]. The most widely involved in early research was the matrix decomposition algorithm based on conditional random fields, but the applicability of the matrix decomposition method was limited by its direct access to prediction scores through vector inner products [15]. With the gradual development of deep learning, the use of neural networks to learn the nonlinear interactions between users and items has become a new hotspot for research [22]. Many recommendation models based on deep learning techniques have been proposed, such as Wide&Deep [10], DeepFM [23], etc. Among them, NCF [24] combines linear matrix decomposition and neural networks to characterize the implicit user–item interaction. Compared with the traditional matrix decomposition method, although this method achieves a great improvement in recommendation performance, it has limitations as it cannot combine item relationships for reasoning. In contrast, reinforcement learning has the ability to handle large-scale data and extract the underlying features. The recommendation model based on deep reinforcement learning can more appropriately adjust and feed back on the recommendation strategy to users through reward and punishment strategies. To address the problems of the non-adaptive propagation and non-robustness of graph neural networks (GNNs) in recommender systems, Fan [25] proposed the graph trend filtering networks for recommendations (GTN), which can capture the adaptive reliability of the interactions. Jiang [26] et al. constructed a retrieval and ranking model that synthesizes information from mathematical expressions with relevant text, extracts ontological attributes from the scientific literature, and further ranks the retrieval results. Gayar [27] et al. proposed an integrated search engine framework that combines the advantages of keyword-based and semantic-ontology-based search engines and solves the problems of the retrieval process, such as unclear retrieval features and a short response time. Kachun et al. [28] proposed a quad-channel graph model (X-2ch) for knowledge embedding, which distills KG information and embeds it as edge attributes in a bi-directional manner to model the natural user–item interaction process, to holistically capture the interconnectivity of users and items while preserving their distinct properties. Yan [29] significantly improved the recommendation accuracy by using gated recurrent units and collaborative filtering algorithms to model users' long- and short-term interests. Although the recommendation system based on reinforcement learning has largely improved the recommendation performance, it ignores the impact of state vectors on the model performance and lacks inter-item correlation analysis.

The approach based on knowledge graphs and deep learning to achieve retrieval recommendations is to perceive and integrate auxiliary information such as entity relationships and item representations [17]. In knowledge-graph-oriented models during training, the designed algorithms mostly implement the intention prediction process in stages, mainly including models for the extraction of graph features and models for the prediction of links. Early research on knowledge graphs focused on path-based models

that used various connectivity patterns of entities for recommendation, such as RKGE [30], Hete-MF [31], and HeteRec [32]. Later, with the development of deep learning, embedding-based approaches emerged. By this end-to-end modeling and learning approach, the graph structure data can be effectively embedded, thus improving the link prediction results. Especially in the field of electricity, the topology of the grid is important for data retrieval; thus, Wang [33] et al. used graph neural networks to mine the connections between entities. In the field of film and television, Yu [34] et al. proposed a graph convolutional network (OR-GCN) based on object relations; this method analyzes directed graphs by building a graph convolutional network and achieves better results in film classification. However, the current embedding-based approach does not consider the connection relationships between information in the knowledge graph, which leads to the weak interpretability of the recommendation system and low user trust. The path-based recommendation system addresses this drawback by making recommendations based on the semantic relationships expressed by paths in the knowledge graph. Such models mainly rely on manually designed paths, and later studies are mostly based on embedding approaches to achieve further recommendations via the multi-layer embedding of knowledge graphs, such as CKE [35], DKN [36], and SHINE [37]. However, an excessive number of paths leads to complex models and also makes the computation and system overhead too large, which reduces the training efficiency of the models. Therefore, hybrid recommendation methods based on the above two types combine word embeddings and path information to achieve user feature mining, such as RippleNet [38] and lntentGC [39]. In addition, Zoomer [40], a recommendation method based on GNNs, introduces the concept of the region of interest (RIO) in the recommendation process, which is also effective in mitigating the problem of low recommendation quality due to information overload, which causes the recommendation model to deviate from the intention of a particular user. Although it refines item features and expresses user preferences more richly, it leads to the confusion of relationships in the knowledge graph and faces the same problems of cold starts and model complexity.

## 3. Recommendation Method for Power Knowledge Retrieval Based on GNNs

The power knowledge retrieval recommendation method based on graph neural networks is mainly composed of two parts: power knowledge reasoning based on a graph neural network and power knowledge retrieval and recommendation based on a knowledge subgraph. The former mainly realizes the correlation coefficient metric among target entities through the embedding and aggregation propagation of power entities, and it achieves knowledge inference for graph complementation through relationship type threshold judgment. The latter seeks to achieve intelligent recommendations via the knowledge aggregation of user retrieval behavior features, forming a fusion subgraph and reordering the recommendation list. The principles of the two techniques are described below.

### 3.1. Power Knowledge Reasoning Based on Graph Neural Network

Compared with traditional deep learning algorithms, graph neural network learning can handle unstructured data like grid topology information and, at the same time, can fit well with the mesh structure of the knowledge graph, making full use of the storage structure of the knowledge graph. Entity feature embedding based on the graph path relationship can obtain a better entity representation neural network (PKR-GNN) for entity embedding and relationship complementation regarding power knowledge graph information. Therefore, this paper proposes a power knowledge reasoning model based on graph neural networks.

3.1.1. Graph Entity Embedding Based on Graph Neural Network

A graph $G = (V, E)$ is defined for the power knowledge graph, where $V$ denotes the set of nodes in the graph and $E$ denotes the set of edges in the graph. In the graph convolutional neural network, the entity embedding phase performs multiple information transfer processes [37]. For a particular node $v$, this is shown in Equation (1):

$$m_v^{t+1} = \sum_{\omega \in N(v)} M_v^t \left( h_v^t, h_\omega^t, e_{v\omega} \right) \tag{1}$$

$m_v^{t+1}$ is the message received by node v at time $t + 1$. $N(v)$ is all the neighboring points of node v. $e_{v\omega}$ is the identity matrix of the edges connecting node v and node w. $M_v^t$ is the message function of node at time t. Equation (1) indicates that the message received by node v originates from the state $h_v^t$ of node v itself and the state $h_\omega^t$ of the neighboring nodes and the edge features connected to it $e_{v\omega}$. The node needs to be updated after generating the message, as shown in Equation (2).

$$h_v^{t+1} = U_v^t \left( h_v^t, m_v^{t+1} \right) \tag{2}$$

$U_v^t(\cdot)$ is the node update function that takes the original node state $h_v^t$ and information $m_v^{t+1}$ as input to obtain the new node state $h_v^{t+1}$. Based on the above approach to node modeling, the edges in the graph can be modeled similarly, with the equation shown in (3):

$$m_{e_{vo}}^{t+1} = M_e^t \left( h_{e_{vo}}^t, h_v^t, h_\omega^t \right) \tag{3}$$

$$h_{e_{v\omega}}^{t+1} = U_e^t \left( h_{e_{v\omega}}^t, m_{e_{v\omega}}^{t+1} \right) \tag{4}$$

$M_e^t$ and $U_e^t$ in Equation (3) are the message transfer function and the state update function for the edge at time $t$.

In the read phase, the whole graph-based feature vector is computed using the read function $R(\cdot)$, which is calculated as shown in Equation (5):

$$\hat{y} = R\left( \left\{ h_v^{\mathrm{T}} \mid v \in G \right\} \right) \tag{5}$$

$\hat{y}$ is the final output vector in the equation. $R(\cdot)$ is the read function that reads the graph embedding vector from the last layer of hidden states of the entire graph node. $T$ is the last duration in the message passing phase. The output vector $\hat{y}$ can be used for the subsequent task of estimating the system state. The function $R(\cdot)$ is a fully connected layer with weight parameters derived from network training. The node messaging function in this paper is shown in Equation (6):

$$m_v^{t+1} = W_n^t h_v^t + \sum_{\omega \in N(v)} h_\omega^t f^t \left( h_{v\omega}^t \right) \tag{6}$$

$W_n$ in Equation (6) is the learnable parameter matrix. $f$ is the activation function of the fully connected layer. The message transfer function of the edge is shown in Equation (7):

$$m_{e_{v\omega}}^{t+1} = W_e^t h_{e_{v\omega}}^t + f^t \left( h_v^t, h_\omega^t \right) \tag{7}$$

$W_e^t$ in the above equation is the learnable parameter matrix. The state update function in this paper is implemented using a gated cyclic unit, as shown in Equations (8) and (9):

$$h_v^{t+1} = \mathrm{GRU}\left( h_v^t, m_v^{t+1} \right) \tag{8}$$

$$h_{e_{v\omega}}^{t+1} = \mathrm{GRU}\left( h_{e_{v\omega}}^t, m_{e_{v\omega}}^{t+1} \right) \tag{9}$$

In summary, the knowledge embedding process of electric power entities based on graph neural networks is shown in Figure 1.

### 3.1.2. Knowledge Reasoning Incorporating Electricity Mapping Paths

In order to realize the potential value of information mining of the power knowledge graph and then improve the completeness of the knowledge graph, we propose a knowledge inference method based on the power graph paths. The method combines the power entity embedding vector representation to realize the mining and prediction of entity relationships. The framework diagram of knowledge inference incorporating mapping paths is shown in Figure 2.

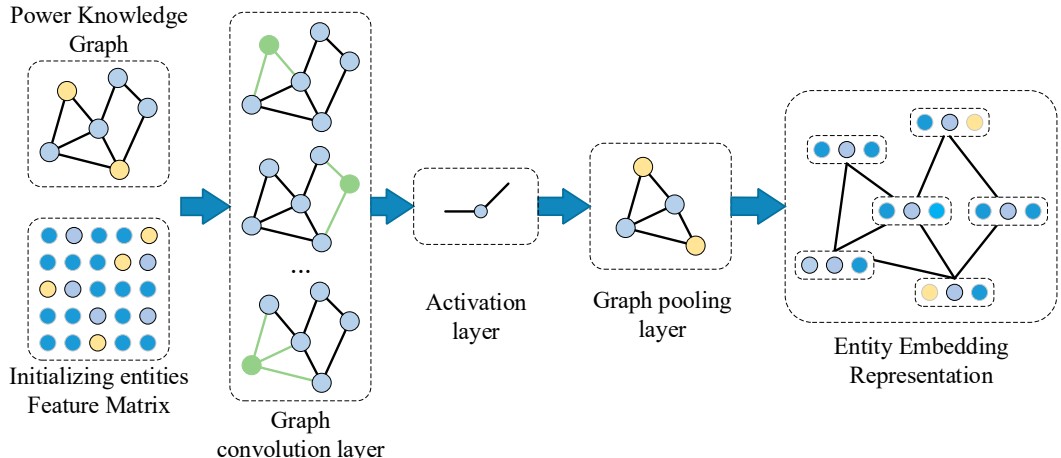

**Figure 1.** Architecture of graph entity embedding based on graph neural network.

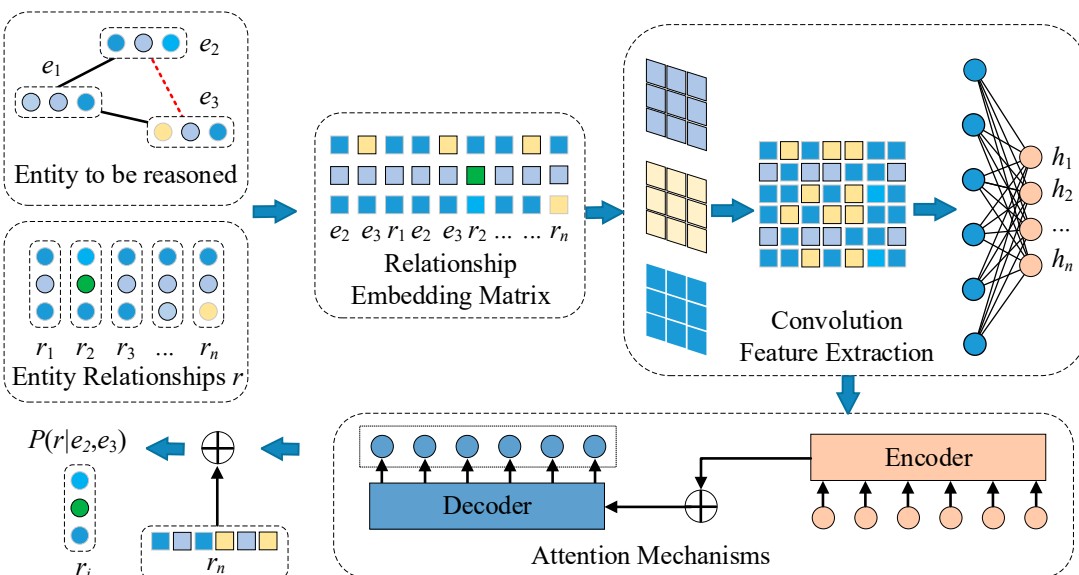

**Figure 2.** Knowledge inference architecture diagram of fused power mapping path.

Step 1: Obtain the entity relationship embedding matrix of the relationship to be evaluated. Based on the power entity embedding vector, we extract the entity relationships to be evaluated in the graph and then obtain the inference entity-related vector representation $(e_1, e_2)$ and relationship list $r_n$. By stitching the entity and relationship list to form the relationship embedding matrix, the path information of the entity to be evaluated can be effectively used to facilitate the selection of a suitable feature extraction model for relationship evaluation.

Step 2: Use a CNN for the feature extraction of the relationship graph. The candidate relations corresponding to head and tail entities are transformed into $k$-dimensional vectors and used for path convolutional coding inputs. The filter size and step size have a large impact on the feature extraction and computational overhead, so a uniform step size and convolution kernel are used to extract features and avoid extracting meaningless local features. We extract all the triads on the path one by one as their local patterns, and we stitch all the features extracted by the convolution kernel to obtain their vector sequence representation $\{h_1, h_2, \cdots, h_n\}$.

Step 3: Score the candidate relations via the attention mechanism. The vector representation $r$ of the candidate relationship is matched with multiple path codes of the entity pair, and the semantic relevance score of each path is calculated and then assigned with independent weights. The state vectors of the candidate relations are weighted and

the probability scores $P(r|e_1, e_2)$ of the candidate relations and the corresponding entity pairs are calculated, which are used to determine whether the triad is valid or not. The calculation process is shown in Equations (10)–(13):

$$\text{score}(p_i, r) = \tanh(p_i W_s) r \tag{10}$$

$$\alpha_i = \frac{\exp(\text{score}(p_\mathbf{i}, r))}{\sum\limits_{i=1}^{n} \exp(\text{score}(p_\mathbf{i}, r))} \tag{11}$$

$$c = \sum_{i=1}^{n} \alpha_i p_i \tag{12}$$

$$P(r|e_s, e_t) = f(W_p(c + r)) \tag{13}$$

$W$ in the above equation is the weight parameter and $f$ denotes the sigmoid activation function. The degree of association of the semantics of candidate relations is measured by weight assignment, and thus different association paths are distinguished.

### 3.2. Power Knowledge Retrieval and Recommendation Based on Knowledge Subgraph

Based on the complementary power knowledge graph, this paper designs the power knowledge retrieval and recommendation method based on a knowledge subgraph (PRR-KS) by combining the features of user retrieval behavior. Knowledge aggregation is performed for different users to form a personality subgraph, and the user retrieval subgraph is analyzed by transforming the embedding of retrieval keywords and matching the similarity of power entity features. Based on the topology structure, the user personality and retrieval subgraphs are fused, and the power entities are reordered based on the subgraph path relationship. Finally, a recommendation list for the target user is generated to achieve the accurate prediction of the user retrieval intention. In the Figure 3, the recommendation architecture of electricity knowledge retrieval based on a knowledge subgraph is shown.

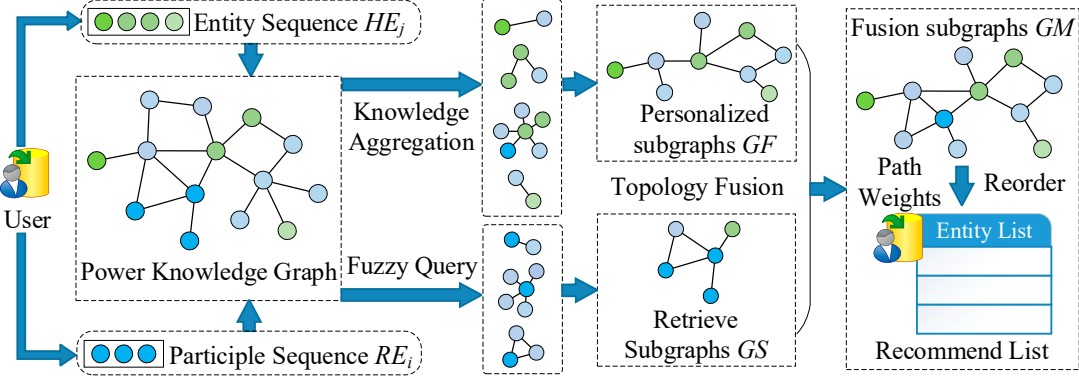

**Figure 3.** Recommendation architecture of power knowledge retrieval based on knowledge subgraph.

Step 1: User behavior feature acquisition and retrieval keyword embedding characterization. For the personalized retrieval features of different users, the entity items selected in the user's historical retrieval records and interface interactions are mapped to the corresponding entities in the electricity knowledge graph. Moreover, the TF-IDF algorithm is used to assign the power entity weights according to the browsing frequency of the items, which is calculated as shown in Equation (14):

$$TF - IDF(x) = \frac{n_{x,j}}{\sum_k n_{k,j}} * \log\left(\frac{N}{N(x) + 1}\right) \tag{14}$$

$n_{x,j}$ denotes the number of occurrences of electricity entity $x$ for the $j$-th user, and $\sum_k n_{k,j}$ denotes the total number of entities for the $j$-th user. $N$ denotes the total number of users, and $N(x)$ denotes the total number of users containing the above electricity entity among all users. The entity sequence $HE_j\{(e_1, w_1), (e_2, w_2), \ldots (e_n, w_n)\}$ is obtained by

calculating the power entity weights of each user. $e_n$ denotes the *j*-th power entity in the retrieved entity sequence of user *n*, and $w_n$ denotes its weight. Since the electricity knowledge graph is processed by entity disambiguation knowledge and other processes during the construction process, it can successfully avoid problems such as multiple meanings of words. Therefore, we use the TF-IDF algorithm as it can cover the user's retrieval features well.

According to the user's retrieval content, we use a fuzzy query for matching in the electric power knowledge graph. Firstly, we segment the user's retrieval content by automatic word segmentation, and, after removing the question words and tone words, we can obtain the sequence of retrieval keywords $SEQ_i(Seq_1, Seq_2, \ldots Seq_n)$. After this, we adopt the regular matching method to match each keyword in the electric power knowledge graph, and the matching principle mainly relies on the consistency and synonymy of the words. Finally, we obtain the corresponding electric power entity $RE_i(re_1, re_2, \ldots, re_n)$ in the electric power knowledge graph, where $re_n$ denotes the electric power entity $Seq_n$ corresponding to the user's keyword $Re_n$ in this retrieval process. The keyword fuzzy query method can maximize the query to the associated entity triad in the knowledge graph and expand the mining of the user's search intent scope.

Step 2: Generation of user personality subgraphs and retrieval subgraphs. The user personality subgraphs are generated by knowledge aggregation for different users' entity sequences $GW_i(e_i) = e_1, e_2, \cdots, e_j \quad (\forall e_i \in HE_j \quad and \quad r < k)$. Firstly, knowledge aggregation is performed on the target entities $e_i$ based on the path relationship r of the power knowledge graph, and the set of user personality entities with a k-hop connection relationship is obtained. After this, all the $GW_i$ in the user entity sequence are fused according to the topological connection relationship, and the TF-ID weight value $w$ corresponding to each entity is stored in the user personality retrieval subgraph $GF\{(e_1, e_2, w_1), (e_2, e_4, w_2), (e_3, e_5, w_3), \cdots, (e_i, e_j, w_n)\}$.

Similarly, the power entity sequence $Re_n$ obtained by the fuzzy query is used to obtain the set of retrieval-associated entities $SW_i(e_i)$ according to the above knowledge aggregation method. Finally, we generate the user retrieval subgraph $GS\{(e_1, e_2, w_1), (e_2, e_4, w_2), (e_3, e_5, w_3), \cdots, (e_i, e_j, w_n)\}$ based on the path connection relationship, which is used to represent the range of user retrieval features. It should be noted that the TF-ID weights of each entity in the retrieval subgraph differ in the calculation process. $n_{x,j}$ denotes the number of occurrences of the electric power entity *x* in the *j*-th retrieval subgraph, $\sum_k n_{k,j}$ denotes the total number of entities in the *j*-th subgraph, $N$ denotes the total number of retrieval subgraphs, and $N(x)$ denotes the total number of subgraphs containing the electric power entity in all retrieval subgraphs.

Step 3: Subgraph fusion and reordering of the recommendation list. We obtain the fused entity list $Res = e_1, e_2, e_3, \cdots, e_n \quad (\forall e \in GS \quad or \quad GF)$ and the fused subgraph $GM\{(e_1, e_2), (e_2, e_4), (e_3, e_5), \ldots, (e_i, e_j)\}$ by merging the subgraphs based on the connection relationship between the user's personality and the retrieved subgraphs. Then, we solve the weights of each entity in the entity list $Res$ based on the user personality subgraph $GF$, user retrieval subgraph $GS$, and fusion subgraph $GM$. They are calculated as shown in Equations (15)–(18):

$$GF(w_i) = \frac{\sum_{j=0}^{n} w(e_i, e_j)}{\sum_n w_n} \quad (e_i, e_j) \in GF \tag{15}$$

$$GS(w_i) = \frac{\sum_{j=0}^{n} w(e_i, e_j)}{\sum_n w_n} \quad (e_i, e_j) \in GS \tag{16}$$

$$GM(w_i) = \frac{count(e_i, r_k)}{Max(count(Res, r_k))} \quad k \in N^+ \tag{17}$$

$$w(res_i) = GM(wi) * (GS(wi) + GF(wi)) \tag{18}$$

Here, *k* denotes the hop size between the nearest neighbor entities in the fusion subgraph, which is used as an influence factor to optimize the relationship weights of

the retrieval subgraph and the personality subgraph. Finally, the weights of each entity in the fused entity list are obtained and ranked according to their sizes, and the entity recommendation list $Res' = \{(e_2, w_2), (e_6, w_6), (e_3, w_3), \cdots, (e_n w_n)\} \quad (\forall e \in GM)$ for users is generated. The process of knowledge subgraph generation and fusion is shown in Figure 4.

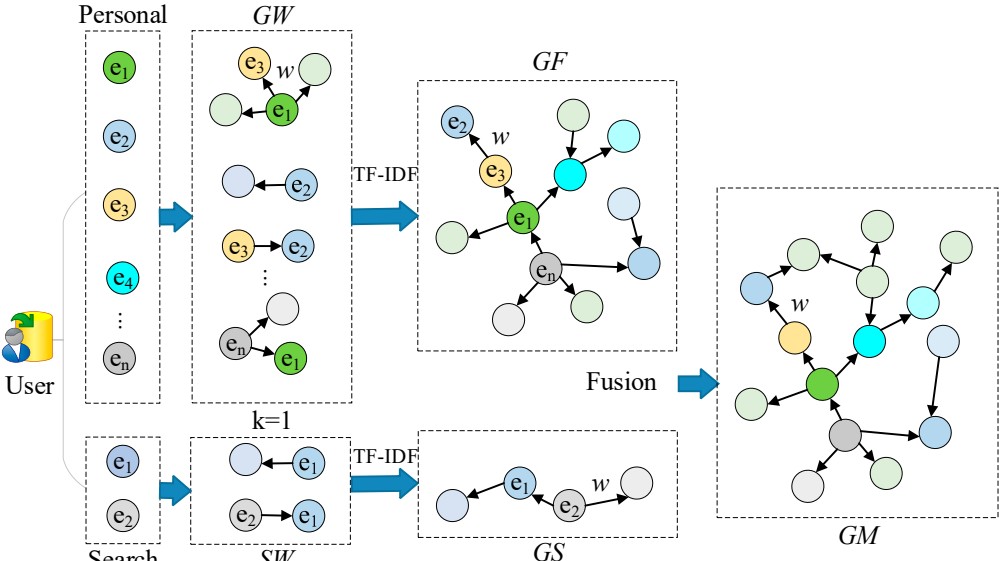

**Figure 4.** Process diagram of subgraph generation and fusion.

## 4. Experimental Analysis

### 4.1. Experimental Datasets

To evaluate the knowledge inference performance of the PKR-GNN model, we use three general-purpose datasets, WN18, WN18RR, and FB15K-237, and one self-built dataset of the power knowledge graph for comparison experiments. The statistical information of the used datasets as shown in Table 1.

**Table 1.** Information on the experimental datasets of knowledge inference.

| Name | Entities | Relationships | Training | Validation | Test |
|------|----------|---------------|----------|------------|------|
| FB15K-237 | 14,541 | 237 | 272,115 | 17,535 | 20,466 |
| WN18 | 40,943 | 18 | 141,442 | 5000 | 5000 |
| WN18RR | 40,943 | 11 | 86,835 | 3034 | 3134 |
| PKG | 1550 | 16 | 4835 | 524 | 506 |

The WN18 dataset is a subset of data from WordNet, a dictionary database that associates English nouns, verbs, adjectives, and adverbs with synonyms. The WN18 dataset is enriched with a large number of symmetric, asymmetric, and reversed relations.

The WN18RR dataset was constructed by eliminating the inverse relations from WN18, in which more symmetric, asymmetric, and combinatorial relations in the original dataset are retained.

The FB15K-237 dataset is extracted from the FreeBase database. This database was mainly constructed by hand and later incorporated some data from Wikipedia, IMDB, Flickr, and other corpora. The relations preserved in this dataset are mainly symmetric, asymmetric, and combinatorial relations, and the inverse relations are also removed.

The power knowledge graph database (PKG) is a knowledge graph based on fault log data, power equipment information, power operation and maintenance manuals, and expert cases in the operation of Southern China's power grid. The recommended method of power knowledge retrieval and the related results obtained in this study will be applied to this knowledge graph.

To evaluate the retrieval recommendation performance of the PKR-GNN model, we conduct comparative experiments using Movie Lens-1M, Lastfm-360K, two general-purpose datasets, and a self-built dataset for power knowledge retrieval.

Movie Lens-1M is the benchmark dataset of the recommender system. Each user in the dataset has attributes, such as gender, age, and occupation.

Lastfm-360K is a music recommendation dataset with data derived from users' rating records from music-on-music websites.

The power knowledge retrieval database (PKR) supplements the power knowledge graph database with historical records of user queries and retrievals.

Detailed information contained in the two publicly available datasets along with a self-constructed dataset is shown in Table 2. During the recommendation experiments, we mainly use these three datasets for comparison.

**Table 2.** Retrieval of information on the recommended experimental datasets.

| Name | Users | Records | Training | Validation | Test |
|---|---|---|---|---|---|
| Movie Lens-1M | 6040 | 1,000,000 | 272,115 | 17,535 | 20,466 |
| Lastfm-360K | 360,000 | 17,000,000 | 141,442 | 5000 | 5000 |
| PKR | 210 | 9600 | 86,835 | 3034 | 3134 |

*4.2. Evaluation Metrics*

The evaluation of knowledge inference models usually scores the correctness of the triad $(e_1, r, e_2)$ and determines whether the triad should be stored in the knowledge graph based on the ranking or a high score. Our proposed PKR-GNN model focuses on the link prediction task, which is to select entities within the set $E$ of candidate entities $eı$ to replace the head entity, i.e., $(eı, r, e)$, $eı \in E$, for each triad $(e_1, r, e_2)$ in the test set and calculate the score $score(eı, e_2)$. Then, all the entities are sorted according to their scores from highest to lowest to find the ranking or position at which the correct triad is located. Correspondingly, the ranking or position of the correct tuple obtained by replacing the tail entity is also calculated. The average value is then taken as the final ranking of this tuple. According to this method, the tuples in the test set are computed sequentially to obtain the sequence of ranking of the test triad positions $\{pos_1, pos_2, \ldots, pos_n\}$. We evaluate all models using three metrics: the mean rank (MR), mean reciprocated rank (MRR), and items with the first K hits (Hit@K). Among them, a smaller the value of the MR metric and a larger value for MRR and Hit@K demonstrate the accuracy of the model's relationship predictions. They are calculated as shown in (19)–(22):

$$MR = \frac{1}{n} \sum_{i=1}^{n} pos_i \tag{19}$$

$$MRR = \frac{1}{n} \sum_{i=1}^{n} \frac{1}{pos_i} \tag{20}$$

$$\text{Hit@}K = \frac{1}{n} \sum_{i=1}^{n} f(pos_i, K) \tag{21}$$

$$f(pos_i, K) = \begin{cases} 1 & pos_i \leq 1 \\ 0 & pos_i > K \end{cases} \tag{22}$$

Search prediction effectiveness evaluation uses the accuracy rate P (precision, P), recall rate R (recall, R), and F1 score (F1) as the evaluation criteria of the model. Among them, the accuracy rate reflects the extent to which the item or entity in the result recommendation list is selected by the user to check, which is calculated as shown in Equation (23). The recall rate reflects the extent to which the item entity that the user chooses to consult in the test set is in the recommendation list provided to the user, which is calculated as shown in Equation (24).

$$P = \frac{TP}{TP + FP} \tag{23}$$

$$R = \frac{TP}{TP + FN} \tag{24}$$

$$F1 = \frac{2PR}{P + R} \tag{25}$$

### 4.3. Experimental Environment and Parameter Settings

In this paper, the word vector dimension of the entity embedding part of the power knowledge map is 200, and each dataset is divided into 100 copies, the experimental epoch number is set to 1000 rounds, and the validation is performed every 100 rounds. The output dimension of the first layer of the convolutional neural network is set to 64, the output dimension of the second layer is set to 32, the batch size is 128, the epoch number is 200, the learning rate is set to 0.001, and the dropout is set to 0.5. We choose the Adam optimizer for training and use the L2 regularization method to prevent overfitting.

The convolutional neural network for the knowledge inference part uses three convolutional layers, one fully connected hidden layer, and one output layer. The dropout value is set to 0.8, the head in the self-attention mechanism is set to 6, and the bias variable is set to 10. In the experiments that involve fusing knowledge subgraphs for retrieval recommendation, the model is optimized by comparatively adjusting the values of the hyperparameters K of the knowledge subgraphs.

The experimental environment is implemented using the deep learning framework PyTorch with CUDA acceleration, and the relevant information is shown in Table 3.

**Table 3.** Experimental environment parameters configuration.

| Environment | Configuration |
|---|---|
| Operating System | Ubuntu 16.04 |
| RAM | 32 GB |
| GPU | GeForce RTX 3070ti $\times$ 2 |
| CPU | Intel Core i7-10700K @ 3.80 GHz |
| Language | Python 3.7 |
| Framework | Pytorch 1.12.0 |

### 4.4. Analysis of Experimental Results

4.4.1. Evaluation of Power Knowledge Reasoning

To fully demonstrate the superiority of the proposed method in this paper, the experiments use two knowledge inference methods, CompGCN and HOLE, as the comparison models. Some of the results are obtained from the original paper, and the models are evaluated according to the evaluation metrics and datasets, respectively.

ConvE [21] is a semantic matching model that extracts semantic information by two-dimensional convolution through multilayer nonlinear operations with high expressive power. This method effectively improves the parameter efficiency and model training speed for multilayer neural networks for knowledge mapping link prediction.

Rotate3D [41] is a knowledge graph embedding model that maps entities into 3D space and defines relationships as rotations from head entities to tail entities. This method is able to naturally maintain the order of relationship combinations by using the non-exchange combination property of rotation in 3D space and performs better in link prediction and path query answering.

DensE [42] performs embedding learning for knowledge graphs based on the translational distance, and it uses a distance-based scoring function to measure the reasonableness of a relational fact as the distance between two entities. The method decomposes complex multiple logical relations and reduces the complexity of the model while preserving its geometric interpretation. It has good performance in complex relational reasoning.

As shown in Table 4, with the comparative evaluation results of power knowledge inference in Figure 5, where the bolded font represents the optimal experimental results

among several models, it is clear that our proposed PKR-GNN method performs better on the FB15K-237 and PKG datasets. The method reduces the MR by 12 and 106 compared to the Rotate3D model, improves the MRR by 0.038 and 0.1, and has the highest score of Hit@1. The reason for this is that the PKR-GNN method not only uses graph convolutional neural networks for the feature extraction of the mapping structure, but also combines attention mechanisms and convolutional neural networks to fully perceive the path and power equipment topology information of entities. Moreover, the inference effect of PKR-GNN is similar to that of the Rotate3D model in both the WN18 and WN18RR datasets. This is due to the fact that the entity relationships within these two types of datasets are more semantic and lexical associations, reducing the advantage of PKR-GNN in fusing path relationships. In contrast to the newer model, the DensE model, the PKR-GNN method has a lower MRR and Hit@1 index and a higher Hit@10 value on the FB15K-237 database. The reason for this may be the fact that this database has more composite relations and therefore performs better in terms of the best inference results, but it affects the accuracy of subsequent relational inference. In summary, our proposed PKR-GNN approach is able to achieve better knowledge inference and performs best in the domain-oriented knowledge inference process.

**Table 4.** Comparative assessment results of power knowledge reasoning.

| Dataset | Model | ConvE | Rotate3D | DensE | PKR-GNN |
|---------|-------|-------|----------|-------|---------|
| WN18 | MR | 374 | **214** | 245 | 231 |
| | MRR | 94.3 | 95.1 | 96.1 | **96.3** |
| | Hit@1 | 93.5 | **94.5** | 94.1 | 94.4 |
| | Hit@10 | 95.6 | 96.1 | 96.3 | **96.5** |
| WN18RR | MR | 4187 | 3328 | 3281 | **3261** |
| | MRR | 43.0 | 48.9 | 47.5 | **47.6** |
| | Hit@1 | 40.0 | **44.2** | 43.2 | 43.5 |
| | Hit@10 | 44.0 | 57.9 | 59.2 | **59.3** |
| FB15K-237 | MR | 244 | 165 | 155 | **153** |
| | MRR | 32.5 | 34.7 | **39.1** | 38.5 |
| | Hit@1 | 23.7 | 25.0 | **27.5** | 27.3 |
| | Hit@10 | 50.1 | 54.3 | 53.2 | **56.3** |
| PKG | MR | 671 | 435 | 517 | **329** |
| | MRR | 83.0 | 79.0 | **90.1** | 89.3 |
| | Hit@1 | 76.2 | 73.6 | 78.3 | **82.7** |
| | Hit@10 | 81.1 | **85.4** | 83.1 | 84.2 |

### 4.4.2. Electricity Knowledge Retrieval Recommendation Evaluation

The selection of the hop count K in the knowledge subgraph has a large influence on the user intention mining in the process of power knowledge retrieval recommendation. Therefore, in order to select suitable parameter values to achieve the accurate prediction of user retrieval intention, this experiment compares different K parameter values. The training effect of the model when K = 1, 2, and 3, respectively, is shown in the figure below.

From Figure 6, it can be seen that our proposed PRR-KS model has a good convergence effect on the training and test sets, and the fluctuations gradually stabilize with the increase in the number of iterations. In particular, the loss value is the smallest and most stable when the number of hops K of the knowledge subgraph is taken as 2. The reason for this is that when the value of K is too small, the effect of recommendation is better than that of K = 3, but there is a limitation in the user's personalized intent perception. This is because, at K = 3, it focuses too much on the user's historical state and ignores the retrieval intent, which increases the interference for the user's retrieval intent matching.

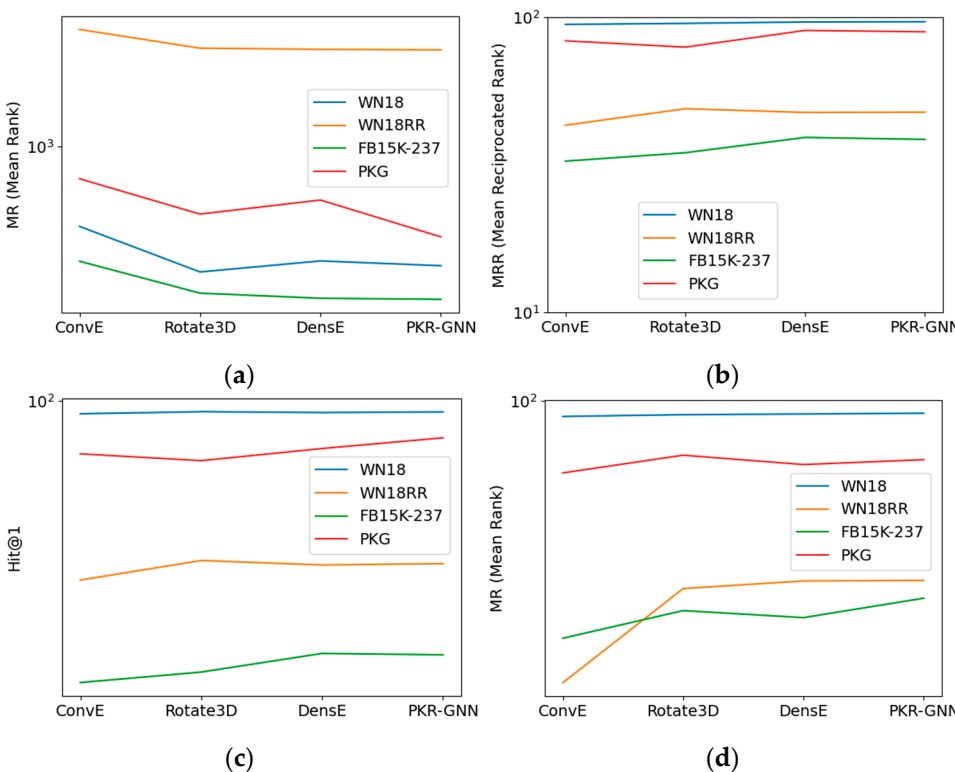

**Figure 5.** Evaluation of knowledge inference effect of each model on different datasets. (**a**) Evaluation results of MR indicator; (**b**) evaluation results of MRR indicator; (**c**) evaluation results of Hit@1 indicator; (**d**) evaluation results of Hit@10 indicator.

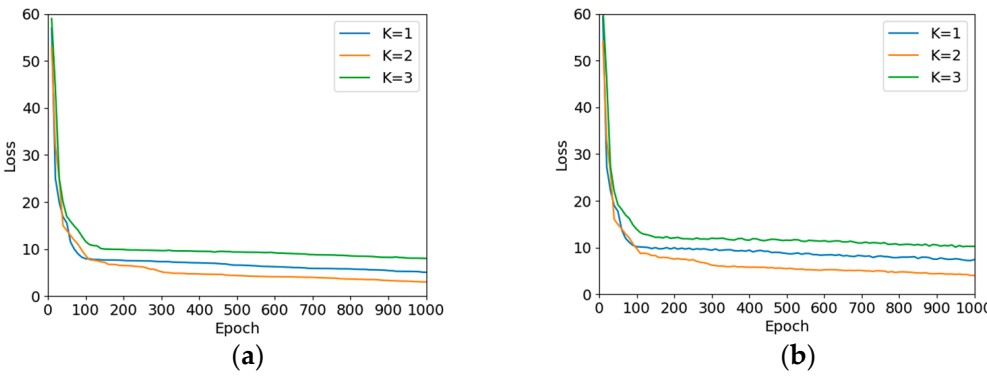

**Figure 6.** Diagram of the training process of the recommendation model for knowledge subgraph retrieval (when K = 1, 2, and 3). (**a**) Training set; (**b**) test set.

In order to fully demonstrate the superiority of the proposed method, the knowledge subgraph at K = 2 is selected for search intention prediction, and two recommendation methods, RippleNet and DeepFM, are used as comparison models. The models are evaluated according to the evaluation metrics and datasets, respectively.

RippleNet [38] is a hybrid recommendation model that propagates user preferences by introducing knowledge graphs, and the inputs are the original features of users and items as well as knowledge graphs.

DeepFM [23] is a deep learning recommendation model that performs the extraction of low-order features and the extraction of high-order features via a neural network module and a factorizer module, respectively, and uses both parts as feature inputs.

KGIN [43] is a knowledge graph recommendation model incorporating graph neural networks, and it designs a new GNN information aggregation mechanism that integrates

sequences of relationships with long-range correlations in a recursive form to enable user behavioral intent prediction.

As shown in Table 5 and Figure 7, where the bolded font represents the optimal experimental results among several models. The PRR-KS method proposed in this paper has the highest F1 value in the comparison experiments of the three datasets, with an average improvement of approximately 7 compared to KGIN, RippleNet, and DeepFM. RippleNet obtains user portraits through path association of the knowledge graph to form a recommendation result ranking, and the accuracy recommendation effect on the Lastfm-360K dataset is similar to that of the PRR-KS method. The PRR-KS method, on the other hand, performs entity embedding and knowledge inference on the knowledge graph through the graph neural network on this basis, and it improves the recall rate by 1.7% compared with the RippleNet model. The recall rate of the PRR-KS method on the Movie Lens-1M dataset is higher than that of the DeepFM model. This is due to the fact that its linear model structure cannot fully exploit the implicit information in the data and ignores the interactions between features, while the PRR-KS method fully takes this into account in the knowledge inference and path weight reconstruction. Comparing PRR-KS with the KGIN model, PRR-KS has a better intention prediction effect, and the evaluation metrics are higher than that of KGIN on both public datasets. However, the R metric in the PRR-KS dataset is slightly poorer, which is supposed to be due to the fact that there are fewer long-distance pairs of entities in this database, and thus its information aggregation is stronger in the process of relevance analysis. In summary, the PRR-KS method proposed in this paper can effectively mine potential knowledge associations and achieve better recommendation results.

**Table 5.** Comparison evaluation results of power knowledge retrieval recommendations.

| Dataset | Model | DeepFM | RippleNet | KGIN | PRR-KS |
|---|---|---|---|---|---|
| Movie Lens-1M | P | **77.1** | 73.6 | 76.8 | 76.5 |
| | R | 73.4 | 71.2 | 72.9 | **74.2** |
| | F1 | 75.2 | 72.4 | 74.8 | **75.3** |
| Lastfm-360K | P | 72.5 | **82.4** | 80.6 | 81.6 |
| | R | 68.2 | 77.9 | 78.7 | **79.6** |
| | F1 | 70.3 | 80.1 | 79.6 | **80.5** |
| PKR | P | 79.3 | 82.2 | 88.1 | **88.3** |
| | R | 71.4 | 80.2 | **86.5** | 86.3 |
| | F1 | 75.1 | 81.2 | 87.2 | **87.3** |

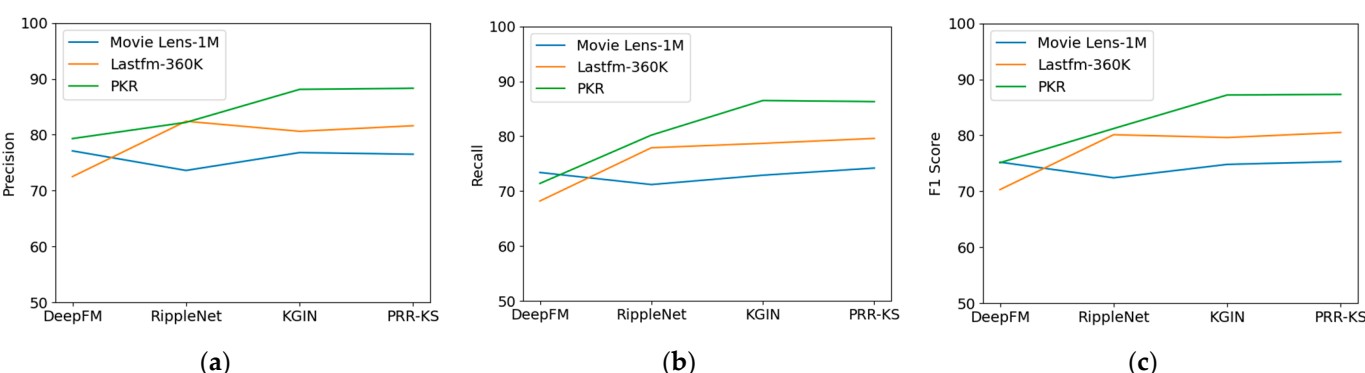

**Figure 7.** Evaluation of the retrieval recommendation effect of each model on different datasets. (**a**) Precision; (**b**) recall; (**c**) F1 score.

The above experiments show that the knowledge inference and retrieval recommendation of PKR-GNN are effective, and it can efficiently and accurately analyze and extract the user's search tendency by combining the power knowledge graph with the user's

personalized data. It can match them to obtain user retrieval features that are closer to the user's search intention.

## 5. Conclusions

To address the problems of grid knowledge and data screening difficulties, we propose a retrieval recommendation method for power fault knowledge based on graph neural networks. Firstly, a graph-neural-network-based power knowledge inference model is constructed to mine the network structure information of the power fault knowledge graph to achieve the deep semantic embedding of power entities and relationships. Secondly, the potential power entity relationships are mined by fusing the power knowledge graph paths, and the power fault knowledge graph is complemented by knowledge inference. Through experimental comparison with the classical models of ConvE and Rotate3D, the experimental results show that the model has a certain generalization ability in knowledge inference and performs better in terms of MR, Hit@10, and other indexes on four datasets. Then, we design a recommendation method for electricity knowledge retrieval based on knowledge subgraphs, which forms a personality subgraph and a user retrieval subgraph through knowledge aggregation. Based on the fusion subgraphs, we realize the electric power entities for reordering and generate recommendation lists for target users to predict the user retrieval intention. By comparison with the classical models of RippleNet and DeepFM, the F1 value of retrieval recommendation on three datasets reaches 87.3, indicating that our method can effectively assist users to achieve the efficient operation and maintenance of power systems.

Although the model has good knowledge inference and recommendation capabilities, due to the domain specialization of the electricity knowledge graph, a large number of expert dictionaries are required and must be relied upon in processes such as keyword matching. Therefore, the model has some limitations when it is applied in other domains. If it needs to be applied to other domains, such as healthcare, the focus should be on improving the generalization ability of the model. Meanwhile, the network structure of this method has a relatively large number of parameters and the model relies on a single training environment. This leads to the problem of slow response times in the inference process of the model, which affects the user's retrieval experience.

The next step will be to investigate adaptive matching for multi-domain retrieval, using either a large model training method or expanding the existing lexicon. On the model side, we will consider reducing the number of network parameters to improve the training efficiency or combining parallel training methods to improve the inference speed of the model and optimizing the parameter settings to improve the convergence speed and stability of the model.

**Author Contributions:** All authors contributed to the writing and revisions; writing—review and editing, R.H.; writing—original draft, Y.Z.; conceptualization, Q.O.; data curation, S.L.; formal analysis, Y.H.; resources, H.W. and Z.Z. All authors have read and agreed to the published version of the manuscript.

**Funding:** This research received no external funding.

**Data Availability Statement:** Data available on request due to privacy or ethical restrictions.

**Conflicts of Interest:** The authors declare no conflict of interest.

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
