# Peer review of "Recommendation Method of Power Knowledge Retrieval Based on Graph Neural Network"

_electronics, doi:10.3390/electronics12183922_

Round 1
Reviewer 1 Report
Novelty:
The paper presents a novel approach to power knowledge retrieval by leveraging Graph Neural Networks (GNNs). Specifically, this approach, termed Recommendation method for Power Knowledge Retrieval based on Graph Neural Network (RPKR-GNN), introduces a new concept of forming a personal subgraph using user retrieval behavior features. The use of a fusion subgraph for reordering entities and generating a recommendation list is novel and has the potential to significantly improve information acquisition and screening in the power grid.
Methods Description:
The methods are adequately described in the paper. The authors have explained how they used graph neural networks to learn the network structure information and achieve the deep semantic embedding. They have also described the process of knowledge inference, fusion of power knowledge graph paths, and the generation of a personal subgraph using user retrieval behavior features.
Comprehensiveness:
The paper provides a thorough exploration of the proposed RPKR-GNN method. It includes a detailed description of the method, the process of semantic embedding, knowledge inference, and user retrieval behavior features. It also presents a comprehensive evaluation of the knowledge inference methods using two models, CompGCN and HOLE, as benchmarks. The evaluation metrics are well explained, and the paper includes a thoughtful discussion on the proposed model's performance compared to the classical models.
Results Presentation:
The results are not clearly presented, with the proposed method being compared against two knowledge inference methods. The evaluation metrics including Mean Rank (MR), Mean Reciprocated Rank (MRR), and items with the first K hits (Hit@K) are not well explained.
Suggestions for Improvement:
- Clarity in Methodology: While the methods are adequately described, there are some areas that could be clarified further. For instance, the process of matching the similarity of retrieval keyword features could be elaborated upon.
- Comparison with Existing Methods: The experimental comparison is done with classical models. However, it would be beneficial if comparisons were also drawn with contemporary models or methods dealing with similar problems. This would provide a better context for the readers to understand the advantages of the proposed method.
- Discussion on Limitations: The conclusion section mentions about improving training efficiency, portability, and generalization ability. However, a separate section discussing the limitations of the proposed method and potential future work could add more depth to the paper.
- Visual Aids: The inclusion of more diagrams or visual aids to illustrate the proposed method, particularly the processes of knowledge inference, fusion of power knowledge graph paths, and generation of a personal subgraph, could enhance the reader's comprehension.
Quality of English Language:
The English language used in the paper is of low quality. The technical concepts are explained in a complex manner. But, the paper's structure is logical, and the flow of ideas is smooth, making it easy for readers to follow the research process and understand the proposed method and its evaluation.
Author Response
Dear Reviewer:
Please see the attachment.
Best regards!

Reviewer 2 Report
Please see the attached reviewer report

Moderate editing of English language required.
Author Response

(The authors gave the same response as above.)

Reviewer 3 Report
The paper introduces a recommendation method for power knowledge retrieval based on Graph Neural Network (GNN) called RPKR-GNN. The proposed method aims to address challenges related to information acquisition and screening in the context of the power grid's digital and intelligent transformation. It utilizes GNNs to learn the network structure of a power fault knowledge graph, perform deep semantic embedding, mine potential power entity relationships, and ultimately generate personalized recommendations for users. The paper claims to outperform various classical models in knowledge inference, Overall, the paper addresses an important issue in the power grid domain and proposes an interesting solution. However, there are some aspects that need attention and improvement:
· High image quality is essential, please provide better quality.
· Discuss the practical implications as well as the future works intentions.
· As the paper presents an important process, it is required that the discussion section should be presented in a clear and separate section to convince readers that the proposed method is more effective than existing studies.
Author Response

(The authors gave the same response as above.)

Reviewer 4 Report
The paper deals with the problems with maintenance technology of power grid which structure and operation is updated very often due to the digital development and its intelligent transformations. Therefore, more and more information is being saved causing problems with their interpretation and finding their important parts. That is why Authors created a certain recommendation method based in Graph Neural Network (RPKR-GNN) to solve this kind of situations.
The paper consists of five parts. The first one is Introduction showing the background to the presented problem. Apart from problem analysis it shows also the explanation why such topic is important and what steps are included in the proposed methodology.
The second part is dedicated to the related work study. Some of it was already done in the first part, but here the amount of references is big enough to say that the topic is well justified. It shows also the recommendation models review what makes the proposed methodology innovative.
The third chapter discusses recommendation method for power knowledge retrieval based on Graph Neural Network. Here, Authors decided to divide it into subparts. First, power knowledge reasoning based on graph neural network is shown, where differences between traditional neural networks and graph neural networks were explained. Also, the mathematical background is provided including graph entity embedding based on Graph Neural Network and Knowledge Reasoning Incorporating Electricity Mapping Paths. Next, power knowledge retrieval and recommendation based on knowledge subgraph is discussed with all necessary mathematical explanations provided.
The content presented in the third chapter was the basis for the experimental analysis which results are presented in chapter four. It includes experimental datasets which were WN18, WN18RR and FB15K-237 as well one self-built dataset of the Power Knowledge Graph for comparison experiments. Next, evaluation matrices are provided allowing to decide whether the certain triad (e1, r, e2) should be stored in the knowledge graph or not. Another part is dedicated to experimental environment and parameter settings. All of these parts allowed to perform the analysis of the results. Again, the Authors did it in subchapters. First, evaluation of power knowledge reasoning was done and then electricity knowledge retrieval recommendation evaluation was performed. It occurred that the proposed solution gave the best results.
The paper ends with conclusions.
In my opinion the paper is ambitious, scientifically sound and the proposed methodology is original and innovative. Therefore, I recommend the paper to be published.
Author Response

(The authors gave the same response as above.)

Reviewer 5 Report
The authors discuss a very interesting topic related to data acquisition using neural networks.
The introduction presented by the authors is satisfactory and shows well the problem and idea presented by the authors and developed later in the article.
The next chapter on related works contains some important items but in my opinion it should be expanded. Authors should refer to more significant literature sources on this subject.
The next part of the article describes quite well how the authors' idea was implemented. Certainly, the key point of the article is the research experiment.
Technical note: In table 4, use the same work z, cannot be xxx once, then 0.xxx, or 0.xx.
I miss discussion. It is true that the authors describe quite extensively what they received in the study, referring to what is in the tables and figures, but this is not a discussion. This should be related, for example, to the works of other authors.
I suggest adding a chapter like this.
Author Response

(The authors gave the same response as above.)

Round 2
Reviewer 1 Report
The proposed method introduces a novel approach for power knowledge retrieval utilizing graph neural networks, which addresses the challenges in information acquisition and screening in the power grid sphere. The fusion of power knowledge graph paths and user retrieval behavior features to form a personal subgraph is an innovative approach. The overall structure and progression of the manuscript are clear and logical. The authors have provided comprehensive details about their methodology and experimental results. The results must be improved. The inclusion of contemporary models DensE and KGIN in the comparison, as suggested in my initial review, provides more context and validates the effectiveness of the proposed method.
The authors have responded adequately to the previous comments and suggestions. The addition of visual aids and diagrams have improved the understanding of the method. However, I would recommend that they further discuss the limitations of their proposed method and potential areas for future work. Also, It would be beneficial to see a more rigorous empirical analysis that compares your work with related studies in the field. I suggest that you consider the following manuscripts for comparison which also deal with similar topics in different contexts:
- https://doi.org/10.3389/fenrg.2020.613331
- https://doi.org/10.3390/e24060810
- https://doi.org/10.1109/ACCESS.2019.2941937
- https://doi.org/10.1016/j.ins.2022.07.034
Consider discussing the similarities and differences in the methodology, the specific challenges each study addresses, and how your research either builds upon or deviates from these works. In addition, a comparative analysis or benchmarking against the approaches used in these papers could strengthen your results and make a more compelling case for the effectiveness of your approach
The English language used in the paper is of moderate quality. The technical concepts are explained in a complex manner. But, the paper's structure is logical, and the flow of ideas is smooth.
Author Response

(The authors gave the same response as above.)

Reviewer 5 Report
thank you, the authors met my demands
Author Response

(The authors gave the same response as above.)
